# How much of the global aerosol optical depth is found in the boundary layer and free troposphere?

Quentin Bourgeois[1,*], Annica M. L. Ekman[1], Jean-Baptiste Renard[2], Radovan Krejci[3],
Abhay Devasthale[4], Frida A.-M. Bender[1], Ilona Riipinen[3], Gwenaël Berthet[2], and Jason L. Tackett[5]

[1]Department of Meteorology (MISU) and Bolin Centre for Climate Research, Stockholm University, Stockholm, Sweden
[2]Laboratoire de Physique et Chimie de l'Environnement et de l'Espace (LPC2E), CNRS/Université d'Orléans, Orléans, France
[3]Department of Environmental Science and Analytical Chemistry (ACES) and Bolin Centre for Climate Research, Stockholm University, Stockholm, Sweden
[4]Atmospheric Remote Sensing Unit, Research and Development Department, Swedish Meteorological and Hydrological Institute (SMHI), Norrköping, Sweden
[5]Science Systems and Applications, Inc., Hampton, Virginia, United States
[*]Now at Federal Office of Meteorology and Climatology MeteoSwiss, Operation Center 1, CH-8058 Zurich-Airport, Switzerland

*Correspondence to:* Quentin Bourgeois (quentin.bourgeois@meteoswiss.ch)

**Abstract.** The global aerosol extinction from the CALIOP space lidar was used to compute aerosol optical depth (AOD) over a nine-year period (2007-2015) and partitioned between the boundary layer (BL) and the free troposphere (FT) using BL heights obtained from the ERA-Interim archive. The results show that the vertical distribution of AOD does not follow the diurnal cycle of the BL but remains similar between day and night highlighting the presence of a residual layer during night. The BL and FT contribute 69% and 31%, respectively, to the global tropospheric AOD during daytime in line with observations obtained in Aire sur l'Adour (France) using the Light Optical Aerosol Counter (LOAC) instrument. The FT AOD contribution is larger in the tropics than at mid-latitudes which indicates that convective transport largely controls the vertical profile of aerosols. Over oceans, the FT AOD contribution is mainly governed by long-range transport of aerosols from emission sources located within neighboring continents. According to the CALIOP aerosol classification, dust and smoke particles are the main aerosol types transported into the FT. Overall, the study shows that the fraction of AOD in the FT - and thus potentially located above low-level clouds - is substantial and deserves more attention when evaluating the radiative effect of aerosols in climate models. More generally, the results have implications for processes determining the overall budgets, sources, sinks and transport of aerosol particles and their description in atmospheric models.

# 1 Introduction

Aerosols influence the radiative budget of the Earth by absorbing or scattering solar radiation and by changing microphysical properties of clouds. Overall, aerosol particles are short-lived (about a week) compounds in the atmosphere due to the efficient removal by dry and wet deposition in the boundary layer (BL) where a majority of the emission sources are located (e.g., Stier et al., 2005; Dentener et al., 2006). As a consequence, most of the aerosol mass is expected to be found in the BL and the aerosol optical depth (AOD), i.e. the aerosol extinction integrated over the atmospheric column, is expected to be dominated by the BL contribution. Indeed, AOD roughly corresponds to aerosol mass concentration because particles need to be optically active (larger than the wavelength of the incident light) to scatter light. Furthermore, if most of the aerosol mass is found within the BL, then BL aerosols might be expected to have the largest climate effect. However, aerosols advected to the free troposphere (FT) have a much longer residence time (typically a few weeks) than those in the BL, potentially inducing a more long-term effect on climate and even acting as a source for particles back to the BL. Light-absorbing aerosols may in addition cause enhanced absorption, and thereby a climate warming effect, if they are located above low-level reflective clouds (Zarzycki and Bond, 2010; Zhang et al., 2014), and they may thus counteract the radiative effect of Twomey cloud brightening (Bender et al., 2016). Light-absorbing aerosols above clouds may also modify cloud formation and evolution, e.g. delay the transition of stratocumulus to cumulus clouds (Yamaguchi et al., 2015; Wilcox et al., 2016). Therefore, accurately characterizing the vertical and regional distribution of aerosols - and in particular discriminating between particles in the BL and FT - is crucial in order to properly assess the role of aerosols in the climate system. Furthermore, understanding the transport processes between BL and FT is a necessity for constraining the overall sources, sinks and budgets of atmospheric aerosol constituents.

The BL is the atmospheric layer directly influenced by the underlying surface and it can in general be assumed to be chemically well-mixed. The BL is governed by a strong diurnal cycle over land and adjusts to surface forcings within a timescale of an hour (Stull, 1988). The continental BL is typically stable and shallow during the night ($<$500 m), and becomes unstable during the day, mixing with the air masses above and extending up to a few kilometers. Processes within the BL control exchange of momentum, heat, moisture and trace gases between the surface and the FT. In addition, the height of the BL - together with the amount of emitted particles - influences the concentration of aerosols near the surface, and consequently governs air quality (Liu et al., 2013; Yang et al., 2015). During severe pollution events, a high concentration of particles may also enhance the stability of the BL, reducing its thickness and further increasing the concentration of particles within the BL (Petäjä et al., 2016). Kafle and Coulter (2013) found evidence that the vertical distribution of AOD does not seem to vary with the diurnal cycle of the BL evolution. Although some recent studies have looked at the vertical distribution of aerosols (e.g., Koffi et al., 2012, 2016; Toth et al., 2016), none has to our knowledge specifically determined the partitioning of AOD between the BL and FT on a global scale.

The parameterization of vertical transport of aerosols in models is crucial for the residence time, long-range transport, dynamic processes and the radiative effect of aerosols (e.g., Textor et al., 2006). Nevertheless, large uncertainties still remain regarding the processes controlling the vertical transport. Several studies have pointed out convective transport and in-cloud scavenging as the main processes determining the vertical distribution of aerosols (e.g., Cui and Carslaw, 2006; Ekman et al.,

2006; Bourgeois and Bey, 2011; Weigel et al., 2011; Kipling et al., 2016). However, other processes such as condensation, biomass burning injection height, deposition and BL mixing have also been reported as important (Kipling et al., 2016; Peers et al., 2016). In addition, pyro-convection and orographic lifting are two regional processes that can transport aerosols from the surface to the FT (e.g., Fromm et al., 2006; Yumimoto et al., 2009; Bourgeois et al., 2015). Although Veira et al. (2015)

showed that only a small fraction (4-5%) of biomass burning plumes reaches the FT, Devasthale and Thomas (2011) indicated that smoke plumes are frequently found above clouds over the ocean near intense biomass burning regions. Peers et al. (2016) reported that many state-of-the-art aerosol-climate models failed to reproduce an observed event where a large smoke plume was transported above clouds over the Eastern tropical Atlantic Ocean. It is thus essential to better constrain atmospheric models in terms of AOD below and above clouds in order to accurately predict the radiative effect of aerosols.

Although atmospheric aerosols have been the subject of intense research for the last decades, aerosol observations covering the whole atmospheric column are still relatively sparse. To fill this gap in knowledge and data availability, the Cloud-Aerosol Lidar with Orthogonal Polarization (CALIOP) satellite instrument, dedicated to measuring vertically resolved attenuated backscatter, was launched in space in June 2006 (Winker et al., 2009). In this study, CALIOP observations are used together with the BL height product from the ERA-Interim archive (Berrisford et al., 2011; Dee et al., 2011) in order to

discriminate the amount of AOD present in the BL and FT (section 2). The main objective of this study is to evaluate the vertical distribution of AOD in the BL and FT during day and night, respectively (section 3). Finally, results are discussed and summarized in section 4.

## 2   Data and methods

The partitioning of the AOD in the BL (from the surface to the top of the boundary layer) and FT (from the boundary layer top

to the tropopause) has been computed from the CALIOP aerosol extinction retrievals and the BL height from the ERA-Interim archive. The final output is monthly averaged AOD values over a two by two degree grid from January 2007 to December 2015. Results for the Antarctica region are likely spurious and thus excluded because high aerosol values are reported by CALIOP despite that the region is known to be pristine.

### 2.1   CALIOP observations

The CALIOP instrument onboard the Cloud-Aerosol Lidar and Infrared Pathfinder Satellite Observation (CALIPSO) satellite is a two-wavelength polarization lidar (532 and 1064 nm) that provides vertically resolved attenuated backscatter between the surface and 40 km for both daytime and nighttime along its orbit (Winker et al., 2009). Depending on the altitude, profiles are sampled at a vertical resolution varying from 30 to 300 m and a horizontal resolution varying from 333 to 5000 m. Aerosols and clouds are separated with the discrimination algorithm described by Liu et al. (2009). Another algorithm classifies

aerosols into different source types (seven aerosol types: marine, dusty marine, dust, polluted dust, clean continental, polluted continental/smoke and elevated smoke) with different physical characteristics (Omar et al., 2009; Winker, 2016). Each aerosol type is associated with a lidar ratio which allows the calculation of the aerosol extinction from the attenuated backscatter

(Young and Vaughan, 2009). Note that the solar background illumination decreases the signal-to-noise ratio during daytime leading to a better aerosol extinction detection sensitivity during the night. The detection threshold of the aerosol extinction also depends on the nature of the particles. Overall, the CALIOP aerosol extinction dectection sensitivity varies between about 0.01 km$^{-1}$ and 0.07 km$^{-1}$, inducing possible misdetections and resulting in a potential underestimate of the lowest extinction values especially at high latitudes and in the FT (Winker et al., 2013; Rogers et al., 2014; Toth et al., 2018). Although uncertainties remain regarding retrieved aerosol extinction values in the troposphere, CALIOP algorithms have been improved over the years leading to a realistic and representative view of global aerosols (Winker et al., 2013). A comparison of CALIOP data with MODIS retrievals is performed below and with in-situ measurements in Section 3.3.

In this study, we use nine years (2007-2015) of CALIOP Version 4.10 Level 2 532 nm aerosol extinction data and the Atmospheric Volume Description (AVD) product containing Feature Classification Flags. With the AVD feature flag, an aerosol type is indicated for each aerosol layer and each profile bin. Note that although the solar background illumination decreases the signal-to-noise ratio during daytime, both daytime and nighttime aerosol extinction data are here analyzed in a similar manner. To minimize the uncertainties associated with the aerosol extinction, the data were screened following the recommendation by Winker et al. (2013): only aerosol extinction values with a Cloud Aerosol Discrimination confidence inclusive of -100 to -20, with an extinction Quality Control (QC) flag value of 0, 1, 18, or 16 and cloud-free scenes were considered, and negative aerosol extinction values and "clear air" (meaning air without aerosols) below an aerosol layer base lower than 2.46 km were removed. The 2.46 km threshold was chosen not to bias low the aerosol extinction near the surface where the layer detection algorithm sometimes places the aerosol layer base above the surface (Winker et al., 2013). Finally, aerosol extinction profiles are averaged only for ten or more profiles per grid box which excludes 1% of data but leads to more robust statistics. With help of the AVD product, the aerosol type of each aerosol extinction data point is determined before the calculation of the AOD. Therefore, AODs are computed for each of the seven aerosol types considered by CALIOP, for both the BL and FT as further described in the Section 2.2.

This screening procedure leads to the global AOD averages over land and ocean listed in Table 1. Our calculated AOD values are in good agreement with those previously derived using data from CALIOP (Winker et al., 2013) as well as the MODIS-Aqua instrument (Collection 6 (Levy et al., 2013)). The slightly larger values in the present study compared to Winker et al. (2013) are most likely due to the use of V4.10 CALIOP data instead of CALIOP V3.

Overall, more than 1.1M CALIOP vertical profiles are used per month. 55% (45%) of them are retrieved during daytime (nighttime) and 40% (60%) of them are retrieved over land (ocean).

## 2.2 ERA-Interim data

The European Centre for Medium-range Weather Forecasts Re-Analysis-Interim (ERA-Interim) archive is based on data assimilation of observations, in particular from satellites, and numerical weather prediction modelling (Berrisford et al., 2011; Dee et al., 2011). The boundary layer height product is diagnosed using the parcel lifting method (or bulk Richardson method) proposed by Troen and Mahrt (1986). The product has been evaluated and showed good agreement with other methods to determine the BL height (von Engeln and Teixeira, 2013). In this study, the ERA-Interim boundary layer height product was used

at a horizontal resolution of 1 degree and with a 3-hour time step. Since the CALIOP spatial and temporal resolution (5 km and 6 s, respectively) is higher than ERA-Interim, the height of the BL is determined for each CALIOP vertical profile by using the corresponding ERA-Interim grid box and the closest time step. This value is used to separate the BL and FT AOD before averaging AOD values over a 2×2 degree grid. The different spatial and temporal resolutions between CALIOP retrievals and ERA-Interim data likely leads to spatial and temporal sampling errors (Schutgens et al., 2016), however, an increase of the horizontal resolution of ERA-Interim data from 1 degree to 0.5 degree changed our AOD results by only 0.3-0.4%. Since the spatial sampling error does not vary significantly, we used 1 degree horizontal resolution assimilation data for computational efficiency.

As expected, the diurnal cycle of the ERA-Interim BL is more pronounced over land than over ocean. The global and annual average of the BL height varies from about 200 m during night to 1400 m during day above land while it remains around 700 m over ocean both during day and night (Figure 1). Sensitivity tests have been performed on the BL height to evaluate its influence on the AOD partitioning between the BL and FT. They showed that a global increase (decrease) of the BL height by 120 m increases (decreases) the global contribution of the BL AOD by about 5%.

## 2.3   The LOAC instrument

The Light Optical Aerosol Counter (LOAC) instrument retrieves the particle number concentration for different size ranges within 0.2-20 $\mu$m (Renard et al., 2016). Mie scattering theory is used for liquid and transparent particles (refractive index = 1.45), and for solid and absorbing particles (refractive index = 2+0.6i), separately. Aerosol extinction values are computed for these two different particle compounds. They represent the range within which the true aerosol extinction of the particle falls in. Finally, the LOAC aerosol extinction at 532 nm is determined by averaging these two extinction values. Since 2014, meteorological balloon flights with LOAC onboard are performed twice a month between morning and noon from Aire sur l'Adour (43°42'N, 0°16'W), France. In this study, we use LOAC measurements for 23 flights spread out between early 2014 and late 2015. Unfortunately, only one single flight is collocated in space and time with a CALIPSO overpass. Therefore, CALIOP vertical profiles were collected in a 2×2 degree grid box centered on Aire sur l'Adour for the closest day (before or after) of each LOAC flight. Note also that particles found in Aire sur l'Adour are representative of a rural aerosol background because the closest mountain (Pyrenees), big city (Toulouse) and ocean (Atlantic Ocean) are located at about 100 km away. So, the comparison between LOAC in-situ measurements and a large 2×2 degree grid box a few days before or after the LOAC flight is relevant.

## 3   Partitioning of aerosols between the boundary layer and the free troposphere

In this section we evaluate the vertical distribution of aerosols and their partitioning between the BL and FT. Since the BL height displays a strong diurnal cycle over land, we analyze the distribution and type of aerosols in the BL and FT during daytime (section 3.1) and nighttime (section 3.2), separately, with an emphasis on daytime which is characterized by a convective boundary layer.

## 3.1 Daytime analysis

The seasonally averaged global distribution of BL and FT daytime AODs over 9 years (2007-2015) are shown in Figure 2. High values of AOD are observed both in the BL and FT over central (all seasons) and southern (JJA and SON) Africa, the Arabian Peninsula, India and eastern China. This distribution is expected as these regions correspond to the main continental
sources of aerosol mass (Ginoux et al., 2001; Giglio et al., 2013; Janssens-Maenhout et al., 2015). Typical features such as transport of aerosols over the Bay of Bengal and the Indo-Gangetic plain (Höpner et al., 2016) or long-range transport of dust aerosols from the Saharan desert over the Atlantic Ocean (e.g., Generoso et al., 2008) are also observed in the CALIOP data, both in the BL and in the FT. Overall, the global daytime AOD is 0.147 in the troposphere, where 0.102 is found in the BL and 0.045 in the FT (Table 2), corresponding to 69% and 31%, respectively, of the total AOD. Although the global AOD over land
(0.182) and ocean (0.129) are very different, the AOD partitioning between the BL and FT is similar over land (71% and 29%, respectively) and ocean (68% and 32%, respectively).

Figure 3 and 4 show that the largest contribution of the FT to the total AOD is found in the polar regions (>60%) where aerosol extinction values are very low and near the detection limits of the instrument. The height of the BL is however relatively low in the polar latitudes (Figure 1) which could explain the large AOD contribution of the FT at these latitudes. Large FT
contributions to the AOD are also found in the tropics, following the seasonal migration of the Inter Tropical Convergence Zone (ITCZ). The FT contribution to AOD reaches 50% at the ITCZ (meaning that BL and FT contributes equally to the total AOD) while the contribution goes down to 10% at mid-latitudes. Figure 4 also shows that the FT contribution to AOD is larger in the northern mid-latitudes (25%) than in the southern mid-latitudes (13%) which is likely due to the difference in land fraction (and associated emission sources and convective activity) in the two hemispheres. Overall, the distribution indicates
that convection largely influences the vertical profile of aerosols which has been extensively reported in the literature (e.g., Cui and Carslaw, 2006; Ekman et al., 2006; Weigel et al., 2011; Kipling et al., 2016).

Figure 5 shows the global average seasonal variation in AOD in BL and FT over land and ocean. The maximum BL and FT AOD is found in MAM and JJA whereas the minimum occurs in DJF. Note that the last three years of the study (2013-2015) show lower BL AOD over land compared with the earlier years (2007-2012). According to our analysis of CALIOP
observations, this is mostly due to large AOD decreases in Northern Africa and Eastern China during these years which is in agreement with findings by Ridley et al. (2014) and Toth et al. (2016), respectively. As indicated by these studies, the AOD decrease in Northern Africa may be due to a reduction in surface winds over the Saharan dust source regions and the AOD decrease in Eastern China may be due to a decrease in aerosol loading after 2008 Olympic games. Overall, the seasonal cycle of the BL AOD over land is correlated with both the FT AOD over land and ocean with a $R^2$ correlation coefficient of 0.45.
In contrast, no correlation is found with the BL AOD over ocean. This result suggests that FT aerosols found over land are often directly related to local aerosol sources while FT aerosols found over ocean are a result of long-range transport (e.g., long range transport of dust aerosols over the Atlantic Ocean).

As described in Section 2.1, seven different aerosol types are discriminated with help of the AVD flag: marine, dusty marine (mixture of dust and marine particles), dust, polluted dust (meaning dust mixed with smoke or other non-depolarizing particles),

clean continental, polluted continental/smoke and elevated smoke. In the following, clean continental aerosol data are not shown because they only contribute to 0.2% of the global AOD (less than 1% in terms of data fraction). The annual contribution of aerosol types in the BL and FT are summarized in Table 3. It should be highlighted that some of the CALIOP aerosol typing will lean toward the BL or FT by design. For instance, an aerosol layer will be classified as clean marine if the layer base is

5 below 2.5 km, over ocean, strongly scattering and weakly depolarizing. Similarly, the dusty marine classification occurs for layers with bases below 2.5 km over ocean and moderately depolarizing. The elevated smoke classification occurs for layers having tops above 2.5 km whereas layers having similar measured optical properties but have tops below 2.5 km are classified as polluted continental/smoke. In Table 3, a tiny fraction of particles classified as clean marine are found over land because CALIOP data have been averaged over a 2×2 degree grid that can be partly land and ocean (see Data and methods section).

Overall, natural aerosols (marine, dusty marine and dust) contribute about 71% of the global AOD while anthropogenic aerosols (polluted continental/smoke and elevated smoke) contribute about 16%. The 13% remaining aerosols are a mix of natural and anthropogenic aerosols (polluted dust). Natural aerosols contribute more to the BL than FT AOD (75% versus 68%) while anthropogenic particles provide a larger contribution to the FT than BL AOD (14% versus 18%). However, within the natural aerosol category, the AOD contribution of pure dust is smaller in the BL than in the FT (21% versus 31%) while

the AOD contribution of marine particles is much larger in the BL than in the FT (36% versus 20%). Overall, marine particles are the aerosol type that is the least efficiently transported into the FT. Indeed, 80% (20%) of the marine particles are found in the BL (FT). In contrast, 60% (40%) of the dust is found in the BL (FT). Smoke aerosols are also efficiently transported into the FT. Although their emission is much smaller than marine and dust aerosols, smoke particles contribute at least 10% of the FT AOD. This can partly be explained by the location of the emission sources (near the convective regions) and aerosol

properties such as hygroscopicity, which is likely lower for smoke than sulfate-containing pollution (e.g., Carrico et al., 2010). Sea salt particles are hydrophilic and efficient cloud condensation nuclei. They are found in pristine areas with very few other aerosol sources which makes them efficiently activated and removed by marine clouds. As a consequence, sea salt aerosols have a much shorter residence time than dust and smoke (less than a day versus several days) (e.g., Stier et al., 2005) and mostly remain in the BL. In contrast, dust and smoke are often emitted in dry convective regions (over desert and/or during the

dry season) resulting in less efficient scavenging and more efficient vertical transport. It should be noted that while the AOD can provide a rough measure of total particulate mass, particle residence time and cloud interactions depend strongly on the particle size distribution. In agreement with Figure 5, BL and FT AOD are correlated for each aerosol type over land ($R^2 \geq 0.7$ for dust, polluted continental/smoke and elevated smoke) but not correlated over ocean (not shown).

## 3.2   Nighttime analysis

During nighttime, the global contribution to the AOD from the BL and FT is 38% and 62%, respectively. In contrast to daytime, most of the AOD is found in the FT due to the low altitude of the BL height at night (about 200 m over land, see section 2.2). However, if we consider the residual layer height (represented by the BL height of the previous day) instead of the nighttime BL defined by ERA-Interim, we find that the nighttime BL and residual layer both contributes about 62% of the total AOD. This is less than the relative contribution of the daytime BL (69%) to the AOD. However, in terms of absolute values, the

daytime BL, and the nighttime BL and residual layer AOD are similar. This indicates that the larger FT AOD during night than during day is likely due to the lower daytime signal-to-noise ratio of CALIOP measurements which can affect the fidelity of feature detection for weakly scattering layers, resulting in a potential daytime underestimate of the aerosol extinction and AOD in the FT (cf. Section 2). In terms of altitude, the vertical distribution of the AOD is similar during day and night with 58-63% of the global AOD average between 0 and 1 km, 21% between 1 and 2 km, 8-11% between 2 and 3 km, 4-6% between 3 and 4 km, 2% between 4 and 5 km, and less than 1% per km above. This result implies that overall, the vertical distribution of AOD does not follow the diurnal cycle of the BL corroborating the results of Kafle and Coulter (2013), and indicating that a large fraction of the aerosol mass remains in the residual layer during night which may then even act as a source of aerosols to the BL forming during the following day (Sun et al., 2013; Curci et al., 2015). This feature can be explained by the dynamics related to the formation and evolution of the boundary and residual layers: the height of the BL is affected by the size of the largest turbulent eddies and thus the spatial scale of efficient mixing, which is in turn determined by the input of solar energy. As the BL height increases, emissions from the surface are mixed with air from higher altitudes, resulting in a well-mixed BL up to a few kilometers. In contrast, the aerosol concentration does not follow the decreasing BL height in the evening, but instead remains in the less dynamic residual layer, where the removal processes are much slower than within the BL.

## 3.3   Comparison with the LOAC in-situ observations

Several studies showed that low aerosol extinction values are underestimated by CALIOP due to limitations in the detection of weakly scattering aerosols, leading to a potential underestimate of aerosols, particularly in the daytime due to solar noise. Kacenelenbogen et al. (2011) showed a case study in which the CALIOP retrieval underestimates an AOD plume of about 0.05-0.06 measured with other instruments by about 40%. Kim et al. (2017) found that undetected layer AOD accounts for about 0.031 (0.036 during day and 0.025 during night). Toth et al. (2018) concluded that CALIOP screened AOD values (i.e. that do not pass the requirement tests) have a primary mode of 0.03-0.05. According to Rogers et al. (2014), AOD in the FT may be underestimated by 0.02. In terms of aerosol extinction, Winker et al. (2013) indicated that true aerosol extinction values of 0.001 km$^{-1}$ or less are underestimated by CALIOP while Sheridan et al. (2012) showed that 95% of aerosol extinction values larger than 0.05 km$^{-1}$ are detected, 50% of aerosol extinction values lower than 0.02 km$^{-1}$ are undetected and almost none of them are detected below 0.01 km$^{-1}$. Although these studies used CALIOP v3 data, our results with CALIOP v4.1 data do not show a different behaviour. Indeed, the study of Sheridan et al. (2012) has been repeated with v4.1 data and we arrived at the same conclusion with regards to extinction at the limits of CALIOP detection (i.e. an underestimate of the lowest aerosol extinction values). As a consequence, the global statistics of the study are very likely biased low, in particular in the free troposphere, as they are more likely to occur in the FT than in the BL (i.e. further away from sources). However, according to the studies previously cited, CALIOP underestimates the AOD by about 0.03-0.05 in which 0.02 would be due to the FT. This indicates that about half of the CALIOP underestimate is due to the BL and the other half to the FT. Therefore, the AOD contribution of the FT may be underestimated and could reach a global value larger than 31%.

A comparison of the aerosol extinction values derived from CALIOP with the LOAC instrument - an in-situ instrument specifically designed to measure low aerosol concentration values - may give additional insights about the potential underesti-

mate of low aerosol extinction values. Figure 6 shows the mean vertical profiles of the aerosol extinction over Aire sur l'Adour for the 23 LOAC flights and for CALIPSO orbit tracks passing in the 2×2 degree grid box centered on Aire sur l'Adour a few days before or after each LOAC flight (because there is only one single LOAC flight collocated in space and time with a CALIPSO overpass). LOAC and CALIOP vertical profiles are in agreement from the surface to about 3-4 km while the aerosol extinction lower than 0.01-0.02 km$^{-1}$ occuring above 3-4 km is slightly underestimated by CALIOP. Considering two different particle types and the 23 flights separately, the fraction of AOD retrieved by LOAC in the FT is between 17% and 31%. In comparison, the fraction of AOD retrieved by CALIOP in the FT is 30%. These results show that CALIOP data agrees well with LOAC observations even though CALIOP data are often underestimated, in particular in the FT.

## 4    Discussion and Conclusion

This paper examines the vertical distribution of the aerosol optical depth (AOD) through its partitioning between the boundary layer (BL) and free troposphere (FT). For this purpose, the CALIOP aerosol extinction product is combined with estimates of BL height from ERA-Interim over a nine year period (2007-2015). The BL and FT contribute with 69% and 31%, respectively, of the global AOD during the day. It should be noted that although CALIOP data have often been reported as underestimated, in particular in the FT, in previous studies due to detection limitations, the case study over Aire sur l'Adour shows a good agreement between the LOAC in-situ measurements and CALIOP data.

The daytime annual average FT contribution ranges from about 10% at mid-latitudes to 50% within the tropics likely indicating a difference in vertical transport efficiency. Convection - occurring more frequently and strongly within the tropics - is often considered as the main transport pathway of chemical compounds from the surface to the FT. This process may be responsible for the difference in the FT contribution to the total AOD between the mid-latitudes and the tropics. The FT contribution to AOD is larger at northern mid-latitudes (25%) than at southern mid-latitudes (13%) which is very likely a consequence of the larger land fraction (and thus emission sources and convective activity) in the Northern than Southern Hemisphere. Continental particles, in particular dust and smoke, are the aerosol types transported the most efficiently to the FT (>40%) while marine particles mostly remain in the marine BL (80%). This implies that continental aerosols in the FT can be transported over long distances towards remote/pristine regions where they can act as a source of aerosols for the BL (Clarke et al., 2013; Bourgeois et al., 2015), impacting the climate far away from their sources.

In contrast to daytime, the BL and FT contribute with 38% and 62%, respectively, of the global AOD during the night. However, when considering the residual layer of the previous day, and including the residual layer in the BL contribution, the contribution of the BL and FT to the total AOD becomes 62% and 38%, respectively. This indicates that a large fraction of the aerosol mass remains in the residual layer during the night (24% in terms of AOD) which may then act as a source of aerosols to the BL forming the following day.

To conclude, this paper shows that a large fraction of the global AOD is found in the FT. This result may be crucial for the calculation of the radiative forcing at the top of the atmosphere in global climate models. Therefore, it is necessary to better constrain models in terms of AOD in the BL and FT. Furthermore, the distribution of aerosol particles between BL and FT

affects their atmospheric residence time, and thus the overall budgets of various particulate phase pollutants. In other words, the results presented here have potential implications not only for calculating the radiative impact of aerosols but also for modeling of atmospheric chemistry, pollutant transport, air quality and aerosol-cloud interactions.

*Code availability.* All codes that have contributed to the results reported in this study are available on request.

5   *Data availability.* CALIOP data have been retrieved through the ICARE Data Services and Center (http://www.icare.univ-lille1.fr) and the ERA-Interim Archive is available on the ECMWF website (ecmwf.int/en/research/climate-reanalysis/era-interim).

*Competing interests.* The authors declare no competing financial interests.

*Acknowledgements.* This work was supported by the Swedish National Space Board (Rymdstyrelsen). We acknowledge ECMWF for providing access to the ERA-Interim Archive (ecmwf.int/en/research/climate-reanalysis/era-interim), and the ICARE Data Services and Center
10   (icare.univ-lille1.fr) for providing access to the CALIOP observations and tools to process them.

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

| | AOD (550 nm) MODIS-Aqua C6 2007-2015 | AOD (532 nm) CALIOP V3 2006-2011 (Winker et al., 2013) | | AOD (532 nm) CALIOP V4.10 2007-2015 *This study* | |
|---|---|---|---|---|---|
| | Day | Day | Night | Day | Night |
| Global ocean | 0.161 | 0.093 | 0.087 | 0.129 (9) | 0.144 (13) |
| Global land | 0.231 | 0.180 | 0.210 | 0.182 (26) | 0.259 (67) |

**Table 1.** Global AOD averages from MODIS-Aqua, CALIOP (Winker et al., 2013) and this study. The $10^{-3}$ annual standard deviation is reported in brackets.

|  | Land | | Ocean | | Global | |
|---|---|---|---|---|---|---|
|  | BL | FT | BL | FT | BL | FT |
| DJF | 0.113 (71) | 0.046 (29) | 0.093 (71) | 0.038 (29) | 0.100 (71) | 0.040 (29) |
| MAM | 0.139 (73) | 0.050 (27) | 0.086 (65) | 0.045 (35) | 0.103 (69) | 0.047 (31) |
| JJA | 0.138 (68) | 0.065 (32) | 0.086 (64) | 0.048 (36) | 0.104 (66) | 0.054 (34) |
| SON | 0.126 (72) | 0.050 (28) | 0.085 (70) | 0.036 (30) | 0.098 (71) | 0.040 (29) |
| Year | 0.129 (71) | 0.053 (29) | 0.087 (68) | 0.042 (32) | 0.102 (69) | 0.045 (31) |

**Table 2.** Seasonal and annual daytime BL and FT AOD average over 9 years (2007 to 2015) of CALIOP data. Standard deviation is smaller than 0.01. Numbers in brackets show the contribution percentages of AOD.

| Aerosol compound | Contribution to total BL AOD (%) | | | Contribution to total FT AOD (%) | | | Fraction present in the FT (%) | | |
|---|---|---|---|---|---|---|---|---|---|
| | Land | Ocean | Global | Land | Ocean | Global | Land | Ocean | Global |
| Marine | 2 | 61 | 36 | 2 | 32 | 20 | 28 | 20 | 20 |
| Dusty marine | 2 | 29 | 18 | 4 | 25 | 17 | 40 | 29 | 30 |
| Dust | 43 | 5 | 21 | 38 | 27 | 31 | 27 | 69 | 40 |
| Polluted dust | 26 | 0 | 11 | 29 | 4 | 14 | 32 | 91 | 36 |
| Polluted continental/smoke | 22 | 4 | 11 | 14 | 3 | 8 | 22 | 32 | 24 |
| Elevated smoke | 5 | 1 | 3 | 13 | 9 | 10 | 52 | 86 | 65 |

**Table 3.** Annual daytime average contribution of each aerosol type to total AOD in BL and FT, and fraction of each aerosol type present in the FT. Data are retrieved from CALIOP for the 2007-2015 time period. The standard deviation is less than 2%.

(a)

(b)

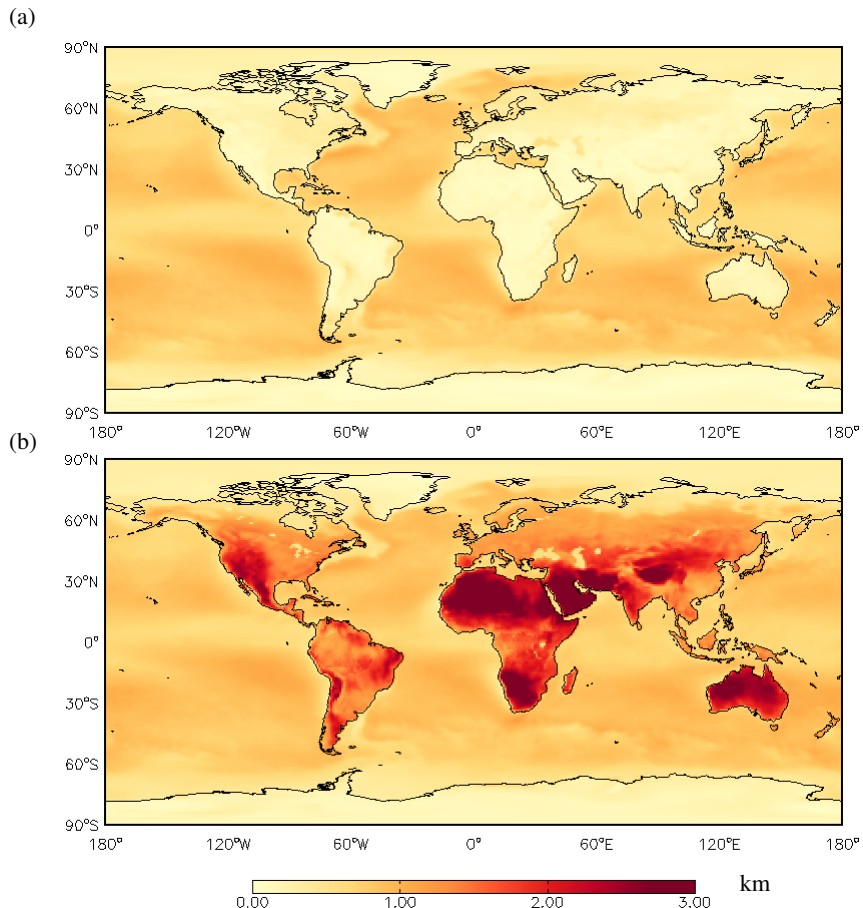

km

**Figure 1.** Annual boundary layer height during nighttime (a) and daytime (b) from ERA-Interim data for the year 2015.

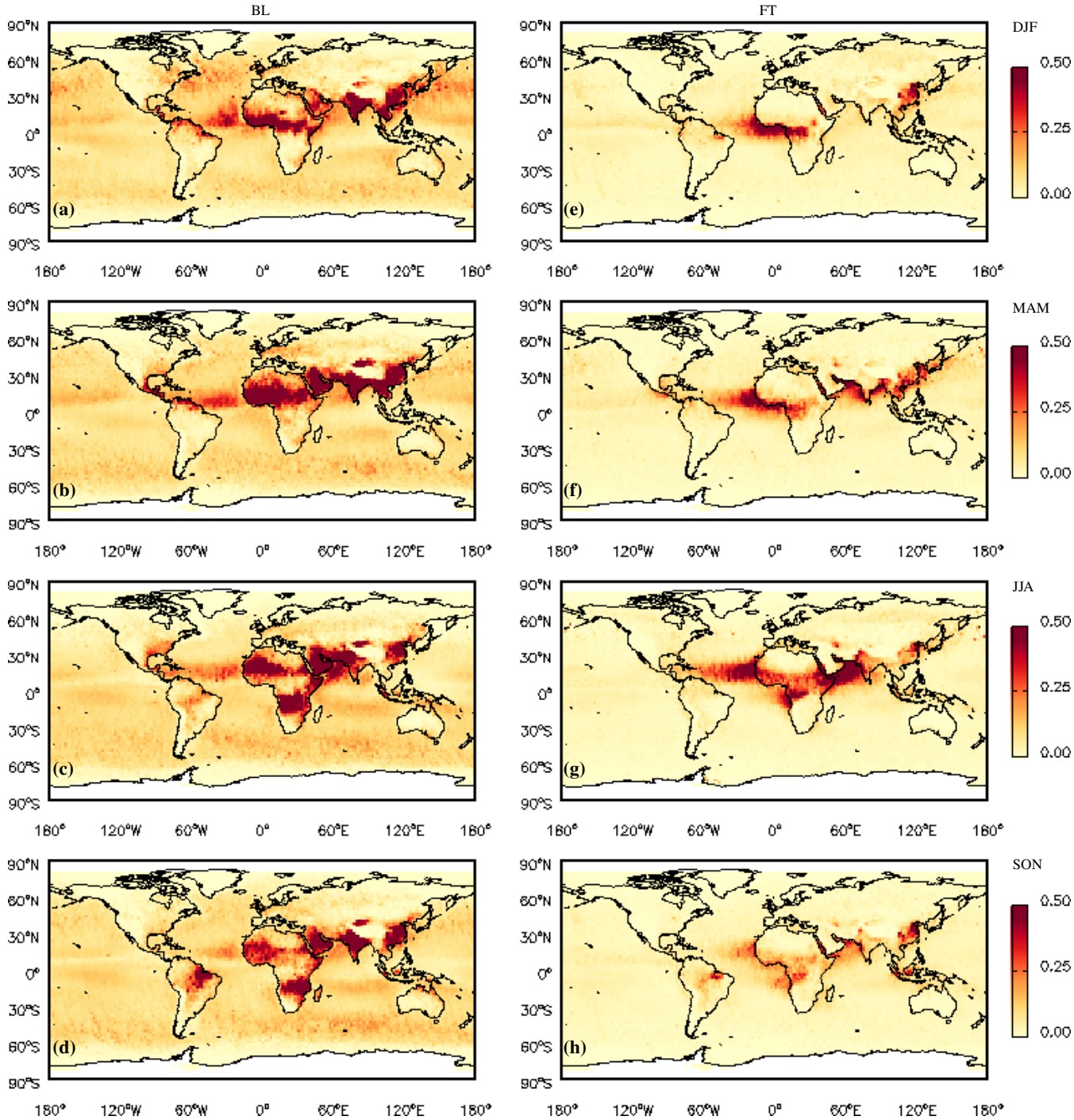

**Figure 2.** Seasonal BL (a: DJF; b: MAM; c: JJA; d: SON) and FT (e: DJF; f: MAM; g: JJA; h: SON) daytime AOD average over 9 years (2007 to 2015) of CALIOP data.

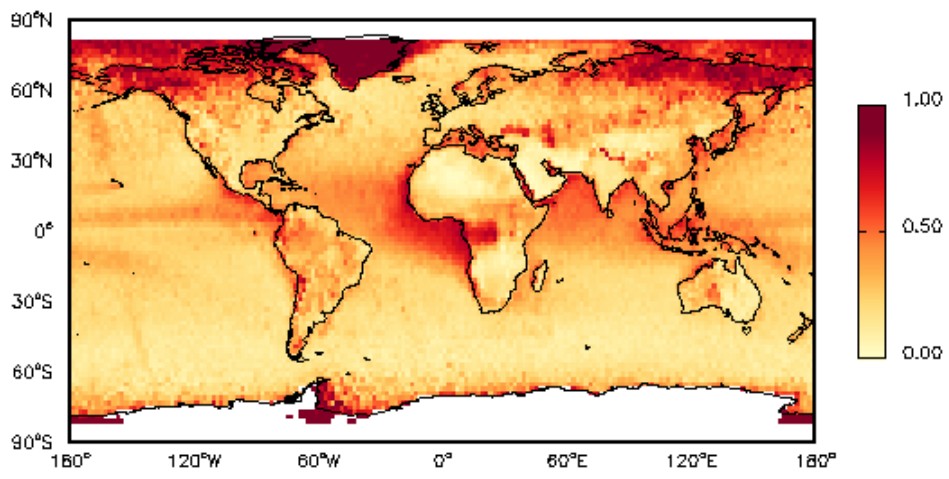

**Figure 3.** Global and annual averaged daytime FT contribution to the total AOD for 9 years (2007 to 2015) of CALIOP data.

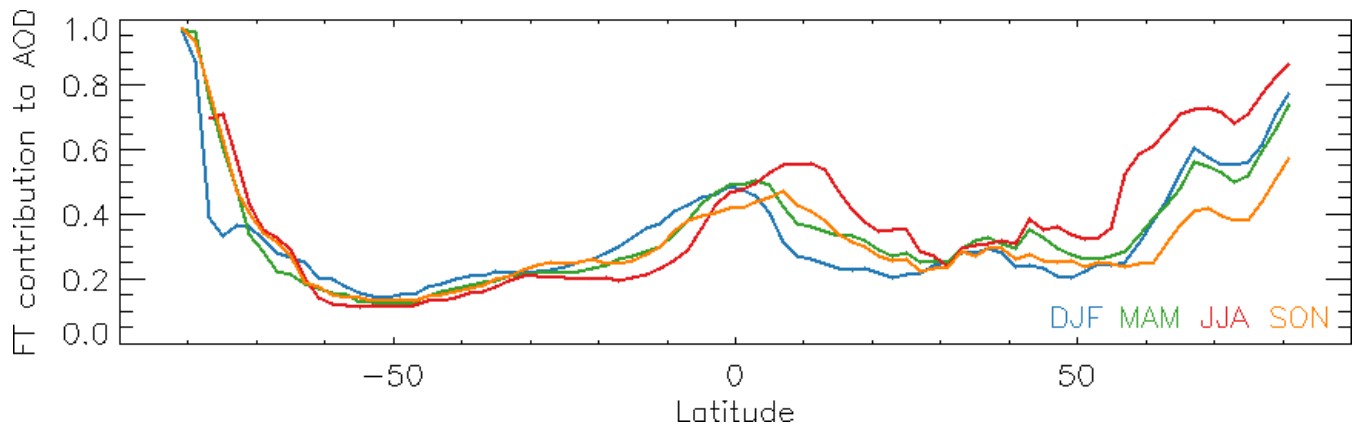

**Figure 4.** Latitudinal distribution of seasonally averaged daytime FT contribution to the total AOD for 9 years (2007 to 2015) of CALIOP data.

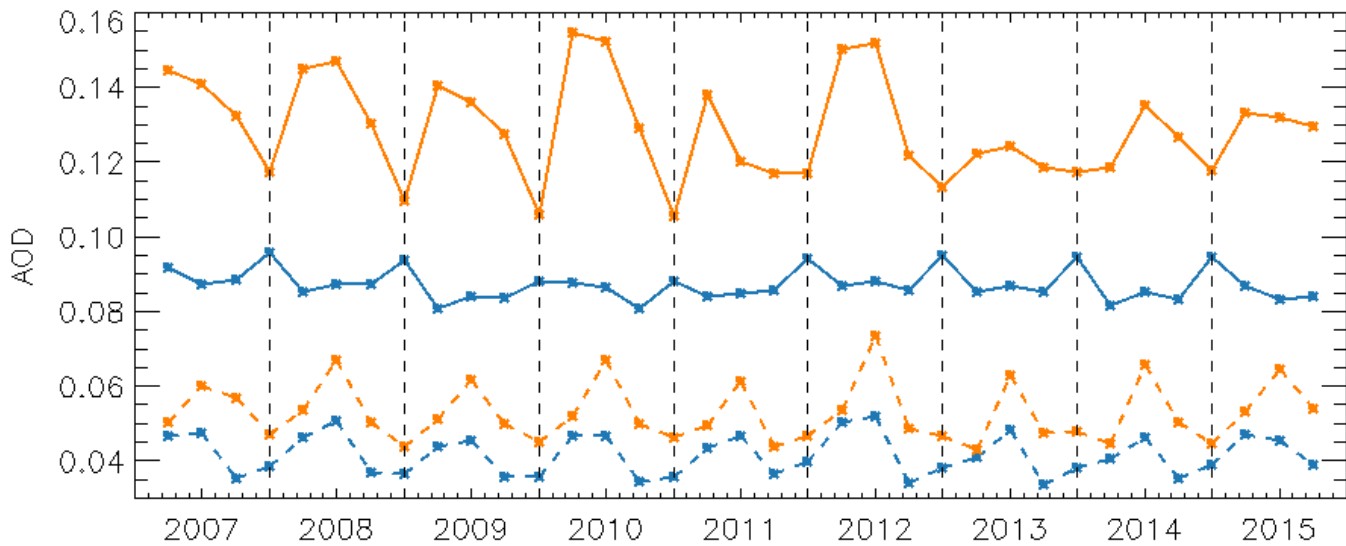

**Figure 5.** Globally and seasonally averaged daytime AOD over land (orange) and ocean (blue) for the BL (solid line) and the FT (dashed line) over 9 years (2007 to 2015) of CALIOP data. Black vertical dashed lines indicate DJF months.

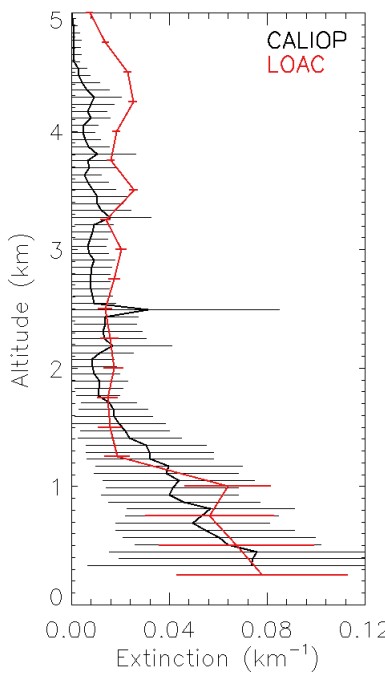

**Figure 6.** Mean aerosol extinction for CALIOP and LOAC over Aire sur l'Adour for 2014-2015 LOAC flights. Horizontal bars show the standard deviation of the CALIOP data and the LOAC aerosol extinction range of the two refractive indices used in the calculation of the aerosol extinction.