# Peer review of "How much of the global aerosol optical depth is found in the boundary layer and free troposphere?"

_Atmospheric Chemistry and Physics, 2017_

## Short Comment (SC1) · 17 Nov 2017

This is a well-written and scientifically important paper.

The authors compare the CALIOP extinction retrievals with 40 profiles derived from a balloon-borne optical particle counter, only one of which was co-located in space and time with the CALIOP profile. There is a much larger data set available for comparison with aircraft flights over Illinois in the USA, where a total of 63 co-located profiles were obtained (Sheridan et al., www.atmos-chem-phys.net/12/11695/2012/, doi:10.5194/acp-12-11695-2012). CALIOP profiles were available for 28 of the 63 co-located aircraft profiles. The conclusion of the comparison reported in Sheridan et

al. (2012) is: "For in situ extinction levels larger than 50 Mmˆ−1 , CALIPSO makes a retrieval 95% of the time (77 out of 81 cases). For the 40–50 Mmˆ−1 extinction bin, CALIPSO's retrieval frequency drops to < 80 %. Below about 20 Mmˆ−1 , CALIPSO detected aerosol extinction only in about 11% (12 out of 109) of the cases, and below 10 Mmˆ−1 , CALIPSO rarely (1 case in 76) retrieved any extinction. For all 244 cases, a 50% probability of detection falls at an in situ extinction level of 20–25 Mmˆ−1."

The inability of CALIOP to detect low levels of aerosol extinction likely leads to a low bias in the retrieved AOD and in the fraction of AOD in the free troposphere. As Andrews et al ( Atmos. Chem. Phys., 17, 6041–6072, 2017; www.atmos-chem-phys.net/17/6041/2017/, doi:10.5194/acp-17-6041-2017) reported, roughly 95% of Earth's surface has AOD below 0.4 (at 440 nm wavelength), and 50% has AOD below about 0.1. If the Sheridan et al. (2012) results, which were based on the CALIOP version 3.01 processing, apply to the Bourgeois et al. results (based on CALIOP version 4.10 processing), then it would seem likely that the global statistics reported by Bourgeois et al. are biased low.

It would be helpful if the authors would include an uncertainty analysis that reflects CALIOP's decreasing probability of detection of aerosol layers as the extinction coefficient drops below 40 Mmˆ-1.

---

## Referee Comment (RC1) · Anonymous Referee #1 · 17 Dec 2017

Review of : How much of the global aerosol optical depth is found in the boundary layer and free troposphere

Quentin Bourgeois, Annica M. L. Ekman, Jean-Baptiste Renard, Radovan Krejci, Abhay Devasthale, Frida A.-M. Bender, Ilona Riipinen, Gwenaël Berthet, and Jason L. Tackett

The authors describe a large set of CALIPSO data used to better estimate the aerosol concentration within the free troposphere (away from sources) and within the boundary layer. This manuscript is of interest for the scientific community but need major revisions before submission to ACP.

[Figure]

MAJOR COMMENTS :

1. The large set of data is always express by years i.e. from 2007-2015. However, it would be useful for the reader to have the numbers of profiles used for each season and each location (Land vs Ocean) and also for each type of particles retrieved by the CALIPSO algorithm. It would increase the confidence for each percentage and value given within this paper.

2. The authors state page 7 that AOD correspond to aerosol mass concentration. From my point of view this statement should be made earlier in the paper to avoid any misunderstanding. As an example, the mass concentration of polluted aerosol is not important while their number concentrations are tremendous.

3. The authors are claiming to compare the CALIPSO data set with airborne in-situ measurements (LOAC). However, the coincident measurements correspond to 1 flight (corresponding to 10 lines in the paper) and therefore are not really relevant for this paper. The CALIPSO data could be compared to ground-based LIDAR measurements all over the world and to in-situ measurements from all the airborne campaigns (such as DISCOVER-AQ, HIPPO, AMMA, etc. . .). This comparison would provide much more information than what one flight with one instrument could provide.

MINOR COMMENTS :

Page 4 – L 12 : What does "clear air" refers to ? I didn't understand why you choose the 2.46km threshold.

Page 5 L 5 : not well said : the influence of the accuracy of the estimated BL height Page 6 L16 AOD over instead of AOD pver.

---

## Referee Comment (RC2) · Anonymous Referee #2 · 18 Dec 2017

This is a concise, well-written paper that addresses a potentially important topic for radiative forcing and atmospheric processes and transport – i.e., the vertical distribution of optically important aerosol. The authors utilize aerosol extinction profiles from CALIPSO and split the aerosol optical depth into boundary layer air and free troposphere air based on boundary layer heights determined from ECMWF ERA interim analysis. It does however need some major revisions in terms of details about uncertainties and the in-situ comparison. I've first provided major science comments and then some minor technical notes and editorial comments.

Major science comments: (1) There is very little space given to uncertainties in the

[Figure]

CALIPSO extinction and AOD retrievals. The authors at some point note that CALIPSO underestimates extinction at values below 0.001 km-1. This should be in the methods section. Additionally, this is actually a pretty high extinction value, corresponding to 100 Mm-1. Most background surface observatories in North America and Europe measure aerosol scattering values less than 0.0005 km-1 (see for example, Pandolfi et al ACPD 2017 and Sherman et al ACP 2015). Scattering tends to be around 90% of extinction (assuming a single scattering albedo of 0.90) so this suggests that CALIPSO retrievals of extinction will also underestimate BL AOD in many locations. The authors give no indication of the magnitude of the underestimation, whether it scales with aerosol loading below 0.001 km-1 or even what the uncertainty is. This is critical information when comparing relative loading of BL and FT.

(2) I appreciate the authors' desire to put the satellite retrievals in the context of in-situ measurements (the LOAC) but feel that this either requires more work or should be removed from the manuscript. The limited nature of the comparison doesn't particularly strengthen the paper and indeed raises more questions than it answers. Some things that should be included if it stays: a) how representative of the 2x2 grid is the region from which the LOAC balloon is launched (it looks like the launch site is close to the Pyranees and the coast and Toulouse which could wreak havoc with the BL height determination and be subject to significant subgrid variability in the aerosol in the BL (and FT) b) provide the size range and assumed refractive indices for converting from size distribution to extinction. c) does the LOAC measure dry aerosol or ambient? If dry what assumptions are made about hygroscopicity to convert to ambient extinction? d) what is the balloon flight path – does it stay in the 2x2 grid around the launch site? If not how does that affect results? e) why not include a plot of the comparisons of the CALIPSO and LOAC profiles? You could show median profiles for the two instruments and use shading to indicate the variability.

(3) I am surprised that the authors did not cite a similar (but better characterized and constrained and with more cases) comparison between CALIPSO and in-situ aerosol

vertical profiles by Sheridan et al in ACP (2012). This paper demonstrates (albeit with an older version of CALIPSO data) the lack of sensitivity of the CALIPSO extinction profiles to extinction values below ∼25 Mm-1 (0.00025 km-1) which is likely larger than the extinction in much of the free troposphere.

(4) Comparisons with ground based and airborne lidar have also explored the FT vs BL loading (e.g., Giannakaki et al (2015) and Rogers et al (2014). The Rogers paper also discusses CALIPSO detection limits (for an earlier version of the data).

Minor comments and editorial notes Page 1 line 5 – need to make clear the limited nature of this comparison in abstract and note location of LOAC flights

Page 1 line 12 – replace process with processes

Page 2 line 24 – replace govern with governs

Page 3 line 20 – some discussion of subgrid variability – a 2x2 degree grid can be pretty variable in terms of aerosol loading – see for example Weigum et al (2016)

Page 3 linen 21 – the Arctic is going to be almost as clean as the Antarctic for the vast majority of its area, particularly in terms of CALIPSOs sensitivity to aerosol extinction. I would suggest some caveats here.

Page 4 line 16-17 – awkward sentence, I'd suggest something like 'AODs are computed for each of the seven aerosol types consider by CALIOP, for both the BL and FT.'

Page 5 lines 9-17 – please see my major comment above. This paragraph needs to have more detail included to make it useful for the reader.

Page 5 lines 16-17 – '. . . particles in the BL and FT are reported as mostly absorbing and scattering, respectively, . . .' I'm not sure what this sentence is supposed to say. In-situ measurements suggest that the aerosol in the BL are primarily scattering – typical single scattering albedo values are ∼0.9 (or higher!) meaning they are ∼90% scattering and 10% absorbing. Sea salt aerosol, which the authors note typically stays

in the BL is pretty much 100% scattering. Please clarify what is meant here.

Page 6 line 3 – put numbers on the FT contribution in polar regions as is done for the other regions further down.

Page 6 line 4 '... detection limits of the instrument.' Please state what these are here and/or in the methods section where the CALIPSO retrievals are described.

Page 6 line 16 – to what does Toth et al 2016 ascribe the AOD decreases observed in Africa and China?

Page 6 line 19 – replace 'indicates' with 'suggests'

Page 6 line 24 – include clean continental aerosol in table 3 and revise this sentence.

Page 6 line 30 – replace 'is' with 'are'

Page 6 line 32,33 – replace 'contribute to about' with 'contribute about'

Page 7 line 5 – replace 'contribute to' with 'contribute'

Page 7 line 7 – the emissions sources will be on the surface (except for airplanes). Do you mean vertical transport or something like that?

Page 7 line 12-13 – rewrite sentence as 'It should be noted that while the AOD can provide a rough measure of total particulate mass, particle residence time and cloud interactions depend strongly on the particle size distribution.'

Page 7 line 20 – replace 'contribute to about' with 'contribute about'

Page 7 line 27 – delete '(not shown)'

Page 8 line 8 – '...scattering or absorbing particles only,...' why would only absorbing particles be considered? The only place those will exist is at the tailpipe of a diesel engine and even there they will have some amount of scattering (SSA∼0.3-0.4).

Page 8 line 11 – delete the sentence about the fraction of number concentration. Since

the LOAC only goes down to ∼0.2 micrometers it's missing a lot (most) of the number concentration.

Page 8 line 19 – See major science note#1. If 0.001 km-1 is where CALIPSO starts underestimating extinction then AOD in much of the BL (except in highly polluted regions) is also going to be underestimated.

Page 8 line 21 – A plot comparing LOAC and CALIPSO for the one coincident profile would be good and a plot showing the statistical comparison with all the LOAC profiles in the 2x2 grid would also be good to give the reader confidence in your results.

Page 9 line 9 – how does the vertical distribution of particles affect their size distribution? Isn't it the other way around?

Page 15 Table 3 – one not include 'clean continental' in the table for completeness?

Page 15 Table 3 – presumably these are based on averages not medians?

Page 17 Figure 2 – it would be interesting to see these maps plotted as the ratio (or difference?) of FT to BL AOD. Doing so would better highlight regional differences. To some extent this is shown in figure 3 but Figure 3 masks the longitudinal differences. For example, is the peak at the equator in figure 3 primarily due to the dust/biomass burning in the 60W to 60E region or does the aerosol get transported outside that longitudinal band and there's actually a FT/BL discrepancy for the full 360? Similarly it would make the Arctic FT contribution more obvious.

Page 19 Figure 4 caption – replace 'full' with 'solid'; replace 'show' with 'indicate'

Cited references:

Giannakaki et al 2015: https://www.atmos-chem-phys.net/15/5429/2015/acp-15-5429-2015.pdf

Rogers et al 2014: https://www.atmos-meas-tech.net/7/4317/2014/amt-7-4317-2014.pdf

Pandolfi et al 2017: https://www.atmos-chem-phys-discuss.net/acp-2017-826/

Sheridan et al 2012: https://www.atmos-chem-phys.net/12/11695/2012/

Sherman et al 2015: https://www.atmos-chem-phys.net/15/12487/2015/acp-15-12487-2015.pdf

Weigum et al 2016: https://www.atmos-chem-phys.net/16/13619/2016/

---

## Author Comment (AC1) · 2 Mar 2018

This is a well-written and scientifically important paper.

The authors compare the CALIOP extinction retrievals with 40 profiles derived from a balloon-borne optical particle counter, only one of which was co-located in space and time with the CALIOP profile. There is a much larger data set available for comparison with aircraft flights over Illinois in the USA, where a total of 63 co-located profiles were obtained (Sheridan et al., www.atmos-chem-phys.net/12/11695/2012/, doi:10.5194/acp-12-11695-2012). CALIOP profiles were available for 28 of the 63 colocated aircraft profiles. The conclusion of the comparison reported in Sheridan et al. (2012) is: "For in situ extinction levels larger than 50 Mmˆ-1 , CALIPSO makes a retrieval 95% of the time (77 out of 81 cases). For the 40–50 Mmˆ-1 extinction bin, CALIPSO's retrieval frequency drops to < 80 %. Below about 20 Mmˆ-1 , CALIPSO detected aerosol extinction only in about 11% (12 out of 109) of the cases, and below 10 Mmˆ-1 , CALIPSO rarely (1 case in 76) retrieved any extinction. For all 244 cases, a 50% probability of detection falls at an in situ extinction level of 20–25 Mmˆ-1."
The inability of CALIOP to detect low levels of aerosol extinction likely leads to a low bias in the retrieved AOD and in the fraction of AOD in the free troposphere. As Andrews et al ( Atmos. Chem. Phys., 17, 6041–6072, 2017; www.atmos-chem-phys.net/17/6041/2017/, doi:10.5194/acp-17-6041-2017) reported, roughly 95% of Earth's surface has AOD below 0.4 (at 440 nm wavelength), and 50% has AOD below about 0.1. If the Sheridan et al. (2012) results, which were based on the CALIOP version 3.01 processing, apply to the Bourgeois et al. results (based on CALIOP version 4.10 processing), then it would seem likely that the global statistics reported by Bourgeois et al. are biased low.
It would be helpful if the authors would include an uncertainty analysis that reflects CALIOP's decreasing probability of detection of aerosol layers as the extinction coefficient drops below 40 Mmˆ-1.

**We would like to thank the referee for these positive words as well as the useful comments.**

**Following the recommendation of the referee, we repeated the calculations of Sheridan et al. [2012] with CALIOP v4.10 data and we arrived at the same conclusion with regards to extinction at the limits of CALIOP detection (i.e. an underestimate of the lowest aerosol extinction values). As a consequence, the global statistics of the study are very likely biased low, in particular in the free troposphere. We expanded the discussion about this matter but we did not add any new figure because it would probably be redundant with Sheridan et al. [2012] results.**

---

## Author Comment (AC2) · 2 Mar 2018

Review of : How much of the global aerosol optical depth is found in the boundary layer and free troposphere

Quentin Bourgeois, Annica M. L. Ekman, Jean-Baptiste Renard, Radovan Krejci, Abhay Devasthale, Frida A.-M. Bender, Ilona Riipinen, Gwenaël Berthet, and Jason L. Tackett

The authors describe a large set of CALIPSO data used to better estimate the aerosol concentration within the free troposphere (away from sources) and within the boundary layer. This manuscript is of interest for the scientific community but need major revisions before submission to ACP.
**We would like to thank the referee for these positive words as well as the careful review. We very much appreciated the suggestions and comments that helped us significantly improve the manuscript.**

MAJOR COMMENTS :

1. The large set of data is always express by years i.e. from 2007-2015. However, it would be useful for the reader to have the numbers of profiles used for each season and each location (Land vs Ocean) and also for each type of particles retrieved by the CALIPSO algorithm. It would increase the confidence for each percentage and value given within this paper.
**Overall, more than 1.1M CALIOP vertical profiles are used per month. 55% (45%) of them are retrieved during daytime (nighttime) and 40% (60%) of them are retrieved over land (ocean). This is now mentioned in the "Data and method / CALIOP observations" section.**

2. The authors state page 7 that AOD correspond to aerosol mass concentration. From my point of view this statement should be made earlier in the paper to avoid any misunderstanding. As an example, the mass concentration of polluted aerosol is not important while their number concentrations are tremendous.
**We agree with the reviewer that the particle number concentration is a primordial parameter for air pollution. However, the focus of the paper is not air pollution but the BL and FT contribution to AOD. Particles need to be optically active (larger than the wavelength of the incident light) to scatter light and thus, to retrieve an AOD. Therefore, the radius - and not the number - of the particles is important and AOD roughly corresponds to the aerosol mass concentration. This is now mentioned in the introduction as well.**

3. The authors are claiming to compare the CALIPSO data set with airborne in-situ measurements (LOAC). However, the coincident measurements correspond to 1 flight (corresponding to 10 lines in the paper) and therefore are not really relevant for this paper. The CALIPSO data could be compared to ground-based LIDAR measurements all over the world and to in-situ measurements from all the airborne campaigns (such as DISCOVER-AQ, HIPPO, AMMA, etc: : :). This comparison would provide much more information than what one flight with one instrument could provide.
**Since there is indeed a single coincident measurement between CALIOP and LOAC, we compare LOAC measurements for 23 flights over Aire sur l'Adour for the 2014-2015 period with CALIOP vertical profiles collected in a 2x2 degree grid box centered on Aire sur l'Adour for the closest day (before or after) of each LOAC flight. Note also that particles found in Aire sur l'Adour are representative of a rural aerosol background because the closest mountain (Pyrenees), big city**

(Toulouse) and ocean (Atlantic Ocean) are located at about 100 km away. So, the comparison between LOAC in-situ measurements and a large 2x2 degree grid box a few days before or after the LOAC flight is relevant. The new Figure 6 shows that the mean vertical profiles of aerosol extinction for LOAC and CALIOP are indeed in agreement, even though CALIOP underestimates low aerosol extinction values in the FT. The underestimate of low aerosol extinction values by CALIOP has also been reported in several other studies that are now mentioned in the LOAC result section. Following the recommendation of another referee, we also repeated the calculations of Sheridan et al. [2012] with CALIOP v4.10 data and 28 collocated flights over Illinois (USA). We arrived at the same conclusion with regards to extinction at the limits of CALIOP detection (i.e. an underestimate of the lowest aerosol extinction values). We expanded the discussion about this matter but we did not add any new figure because it would probably be redundant with Sheridan et al. [2012] results.

MINOR COMMENTS :

Page 4 – L 12 : What does "clear air" refers to ? I didn't understand why you choose the 2.46km threshold.
**"Clear air" refers to air without aerosols. This is now mentioned in the manuscript. We did not "choose" the 2.46km threshold but we used the recommendation of Winker et al. [2013] in order to screen the data properly.**
Page 5 L 5 : not well said : the influence of the accuracy of the estimated BL height
**The sentence has been changed for: "Sensitivity tests have been performed on the BL height to evaluate its influence on the AOD partitioning between the BL and FT.".**
Page 6 L16 AOD over instead of AOD pver.
**Done.**

---

## Author Comment (AC3) · 2 Mar 2018

This is a concise, well-written paper that addresses a potentially important topic for radiative forcing and atmospheric processes and transport – i.e., the vertical distribution of optically important aerosol. The authors utilize aerosol extinction profiles from CALIPSO and split the aerosol optical depth into boundary layer air and free troposphere air based on boundary layer heights determined from ECMWF ERA interim analysis. It does however need some major revisions in terms of details about uncertainties and the in-situ comparison. I've first provided major science comments and then some minor technical notes and editorial comments.
**We would like to thank the referee for these positive words as well as the careful review. We very much appreciated the suggestions and comments that helped us significantly improve the manuscript.**

Major science comments:
(1) There is very little space given to uncertainties in the CALIPSO extinction and AOD retrievals. The authors at some point note that CALPISO underestimates extinction at values below 0.001 km-1. This should be in the methods section. Additionally, this is actually a pretty high extinction value, corresponding to 100 Mm-1. Most background surface observatories in North America and Europe measure aerosol scattering values less than 0.0005 km-1 (see for example, Pandolfi et al ACPD 2017 and Sherman et al ACP 2015). Scattering tends to be around 90% of extinction (assuming a single scattering albedo of 0.90) so this suggests that CALIPSO retrievals of extinction will also underestimate BL AOD in many locations. The authors give no indication of the magnitude of the underestimation, whether it scales with aerosol loading below 0.001 km-1 or even what the uncertainty is. This is critical information when comparing relative loading of BL and FT.
**The CALIOP detection sensitivity is added to the method section. However, we are surprised by the second part of the comment. An extinction of 0.001 km-1 does not correspond to 100 Mm-1 but to 1 Mm-1 (0.001 km-1 => 0.000 001 m-1 => 1 Mm-1). As a consequence, the whole comment is irrelevant because the reviewer supposes that 0.001 km-1 is a large extinction value while it is actually very low.**

(2) I appreciate the authors' desire to put the satellite retrievals in the context of in-situ measurements (the LOAC) but feel that this either requires more work or should be removed from the manuscript. The limited nature of the comparison doesn't particularly strengthen the paper and indeed raises more questions than it answers. Some things that should be included if it stays: a) how representative of the 2x2 grid is the region from which the LOAC balloon is launched (it looks like the launch site is close to the Pyranees and the coast and Toulouse which could wreak havoc with the BL height determination and be subject to significant subgrid variability in the aerosol in the BL (and FT) b) provide the size range and assumed refractive indices for converting from size distribution to extinction. c) does the LOAC measure dry aerosol or ambient? If dry what assumptions are made about hygroscopicity to convert to ambient extinction? d) what is the balloon flight path – does it stay in the 2x2 grid around the launch site? If not how does that affect results? e) why not include a plot of the comparisons of the CALIPSO and LOAC profiles? You could show median profiles for the two instruments and use shading to indicate the variability.

a) **Particles found in Aire sur l'Adour are representative of a rural aerosol background because the closest mountain (Pyrenees), big city (Toulouse) and ocean (Atlantic Ocean) are located at about 100 km away. As a consequence, LOAC remains in this background aerosol**

<ol type="a" start="2">
<li>environment within the first kilometers of the balloon ascension and the choice of a 2x2 degree grid (about 200x200 km) is relevant.</li>
<li>The size range of LOAC is 0.2-20 microns. Typical refractive indices are used: 1.45 for liquid and transparent particles, and 2+0.6i for absorbing particles. This is now mentioned in the method section.</li>
<li>LOAC measures ambient aerosols so no assumptions have been made about hygroscopicity.</li>
<li>As mentioned in the answer a), measurements are always made within the 2x2 degree grid, at least within the troposphere. It is however possible that LOAC flies outside of this grid box in the stratosphere.</li>
<li>A plot showing the mean vertical profile of aerosol extinction values over Aire sur l'Adour for the 23 LOAC flights and for CALIPSO orbit tracks passing in the 2x2 degree grid box centered on Aire sur l'Adour a few days before or after each LOAC flight (because there is only one single LOAC flight collocated in space and time with a CALIPSO overpass) was included (Figure 6).</li>
</ol>

(3) I am surprised that the authors did not cite a similar (but better characterized and constrained and with more cases) comparison between CALIPSO and in-situ aerosol vertical profiles by Sheridan et al in ACP (2012). This paper demonstrates (albeit with an older version of CALIPSO data) the lack of sensitivity of the CALIPSO extinction profiles to extinction values below 25 Mm-1 (0.00025 km-1) which is likely larger than the extinction in much of the free troposphere.

**Same unit conversion mistake, 25 Mm-1 corresponds to 0.025 km-1 (not 0.00025 km-1). However, following the recommendation of the referee, we repeated the calculations of Sheridan et al. [2012] with CALIOP v4.10 data and 28 collocated flights over Illinois (USA). We arrived at the same conclusion with regards to extinction at the limits of CALIOP detection (i.e. an underestimate of the lowest aerosol extinction values). As a consequence, the global statistics of the study are very likely biased low, in particular in the free troposphere. We expanded the discussion about this matter but we did not add any new figure because it would probably be redundant with Sheridan et al. [2012] results.**

(4) Comparisons with ground based and airborne lidar have also explored the FT vs BL loading (e.g., Giannakaki et al (2015) and Rogers et al (2014). The Rogers paper also discusses CALIPSO detection limits (for an earlier version of the data).

**We now include the Rogers et al. [2014] paper in the discussion. However, we cannot really compare Giannakaki et al. [2015] findings with our results because Giannakaki paper uses the PollyXT algorithm for the retrieval of the PBL while we use ECMWF. Note also that the Giannakaki study uses nighttime data only. According to Korhonen et al. [2014], the PBL is found during nighttime at about 1100m in PollyXT vs 200m in ECMWF (see their Figure 10). As a consequence, Giannakaki et al. find a nighttime FT contribution of 46% to the AOD while we find a nighttime FT contribution of 93% (because only the first 200 m are found in the BL in our study).**
**Korhonen et al. [2014]: https://www.atmos-chem-phys.net/14/4263/2014/acp-14-4263-2014.pdf**

Minor comments and editorial notes
Page 1 line 5 – need to make clear the limited nature of this comparison in abstract and note location of LOAC flights
**The location of LOAC flights (Aire sur l'Adour, France) has been added.**
Page 1 line 12 – replace process with processes
**Done.**
Page 2 line 24 – replace govern with governs
**Done.**
Page 3 line 20 – some discussion of subgrid variability – a 2x2 degree grid can be pretty variable in terms of aerosol loading – see for example Weigum et al (2016)

**The effect of subgrid variability would be relevant if a model with a "coarse" resolution was used in the study but we average individual vertical profiles having a horizontal resolution of 5 km on a 2x2 degree grid.**

Page 3 line 21 – the Arctic is going to be almost as clean as the Antarctic for the vast majority of its area, particularly in terms of CALIPSOs sensitivity to aerosol extinction. I would suggest some caveats here.

**We just mention that we do not use CALIOP data over the Antarctic region because they are likely spurious as also reported in Winker et al. [2013]. Indeed, high aerosol values are reported with CALIOP despite it is known to be a pristine region.**

Page 4 line 16-17 – awkward sentence, I'd suggest something like 'AODs are computed for each of the seven aerosol types consider by CALIOP, for both the BL and FT.'

**Done.**

Page 5 lines 9-17 – please see my major comment above. This paragraph needs to have more detail included to make it useful for the reader.

**See answer to the major comment above.**

Page 5 lines 16-17 – ': : : particles in the BL and FT are reported as mostly absorbing and scattering, respectively, : : :' I'm not sure what this sentence is supposed to say. In-situ measurements suggest that the aerosol in the BL are primarily scattering – typical single scattering albedo values are _0.9 (or higher!) meaning they are _90% scattering and 10% absorbing. Sea salt aerosol, which the authors note typically stays in the BL is pretty much 100% scattering. Please clarify what is meant here.

**This sentence has been removed since it was confusing and it has been added earlier that Mie scattering theory is used for two different particles (liquid and transparent particles, refractive index = 1.45 & solid and absorbing particles, refractive index = 2+0.6i).**

Page 6 line 3 – put numbers on the FT contribution in polar regions as is done for the other regions further down.

**Done.**

Page 6 line 4 ': : : detection limits of the instrument.' Please state what these are here and/or in the methods section where the CALIPSO retrievals are described.

**The CALIOP aerosol extinction detection sensitivity is now mentioned in the method section. A range is given because it also depends on the nature of the particles (i.e. the lidar ratio).**

Page 6 line 16 – to what does Toth et al 2016 ascribe the AOD decreases observed in Africa and China?

**Toth et al. [2016] attribute the AOD decrease in China to a decrease in aerosol loading after the 2008 Olympic games and Ridley et al. [2014] attribute the AOD decrease in Northern Africa to a decrease in dust emissions due to a reduction in surface winds over the dust source regions. This is now mentioned.**

Page 6 line 19 – replace 'indicates' with 'suggests'

**Done.**

Page 6 line 24 – include clean continental aerosol in table 3 and revise this sentence.

**Since the contribution of clean continental aerosols to the AOD is almost null, we do not think that it is relevant to include them into table 3.**

Page 6 line 30 – replace 'is' with 'are'

**Done.**

Page 6 line 32,33 – replace 'contribute to about' with 'contribute about'

**Done.**

Page 7 line 5 – replace 'contribute to' with 'contribute'

**Done.**

Page 7 line 7 – the emissions sources will be on the surface (except for airplanes). Do you mean vertical transport or something like that?

**We changed the sentence by "This can partly be explained by the location of the emission sources (near the convective regions)".**

Page 7 line 12-13 – rewrite sentence as 'It should be noted that while the AOD can provide a rough measure of total particulate mass, particle residence time and cloud interactions depend strongly on the particle size distribution.'
**Done.**
Page 7 line 20 – replace 'contribute to about' with 'contribute about'
**Done.**
Page 7 line 27 – delete '(not shown)'
**Done.**
Page 8 line 8 – ': : :scattering or absorbing particles only,: : :' why would only absorbing particles be considered? The only place those will exist is at the tailpipe of a diesel engine and even there they will have some amount of scattering (SSA_0.3-0.4).
**This sentence has been removed since it was confusing.**
Page 8 line 11 – delete the sentence about the fraction of number concentration. Since the LOAC only goes down to _0.2 micrometers it's missing a lot (most) of the number concentration.
The sentence about the aerosol number concentration has been removed.
**Done.**
Page 8 line 19 – See major science note#1. If 0.001 km-1 is where CALIPSO starts underestimating extinction then AOD in much of the BL (except in highly polluted regions) is also going to be underestimated.
**See answer to major comments. An extinction of 0.001 km-1 is very low and the extinction in the BL is usually substantially larger than that.**
Page 8 line 21 – A plot comparing LOAC and CALIPSO for the one coincident profile would be good and a plot showing the statistical comparison with all the LOAC profiles in the 2x2 grid would also be good to give the reader confidence in your results.
**A plot showing the mean vertical profile of aerosol extinction values over Aire sur l'Adour for the 23 LOAC flights and for CALIPSO orbit tracks passing in the 2x2 degree grid box centered on Aire sur l'Adour a few days before or after each LOAC flight (because there is only one single LOAC flight collocated in space and time with a CALIPSO overpass) was added (Figure 6).**
Page 9 line 9 – how does the vertical distribution of particles affect their size distribution? Isn't it the other way around?
**The sentence has been changed for "the distribution of aerosol particles between BL and FT affects their atmospheric residence time".**
Page 15 Table 3 – one not include 'clean continental' in the table for completeness?
**As mentioned above, we do not think that it is relevant to include them into table 3 because their contribution to the AOD is almost null.**
Page 15 Table 3 – presumably these are based on averages not medians?
**The reviewer is right. We added that results in Table 3 are averages.**
Page 17 Figure 2 – it would be interesting to see these maps plotted as the ratio (or difference?) of FT to BL AOD. Doing so would better highlight regional differences. To some extent this is shown in figure 3 but Figure 3 masks the longitudinal differences. For example, is the peak at the equator in figure 3 primarily due to the dust/biomass burning in the 60W to 60E region or does the aerosol get transported outside that longitudinal band and there's actually a FT/BL discrepancy for the full 360? Similarly it would make the Arctic FT contribution more obvious.
**We included a plot (Figure 3) showing the annual and global FT contribution to the AOD. As supposed by the reviewer, the peak at the Equator in Figure 4 is mostly due to the dust/biomass burning in Africa and Indonesia, and the Arctic FT contribution is more obvious.**
Page 19 Figure 4 caption – replace 'full' with 'solid'; replace 'show' with 'indicate'
**Done.**

Cited references:
Giannakaki et al 2015: https://www.atmos-chem-phys.net/15/5429/2015/acp-15-5429-2015.pdf
Rogers et al 2014: https://www.atmos-meas-tech.net/7/4317/2014/amt-7-4317-2014.pdf
Pandolfi et al 2017: https://www.atmos-chem-phys-discuss.net/acp-2017-826/

Sheridan et al 2012: https://www.atmos-chem-phys.net/12/11695/2012/
Sherman et al 2015: https://www.atmos-chem-phys.net/15/12487/2015/acp-15-12487-2015.pdf
Weigum et al 2016: https://www.atmos-chem-phys.net/16/13619/2016/

---

## Author Response (AR2)

Reviewer #2:

I think the authors have responded well to my comments with the one exception listed below.

**We would like to thank the referee for the careful review. We very much appreciated the suggestions and comments that helped us significantly improve the manuscript.**

Page 5 lines 16-18 – it's still unclear to me how the extinction is calculated from the LOAC. The authors provide two different refractive indices (RI) but no indication is given of which RI is used when to determine extinction from the LOAC size distributions. The second RI (2+0.6i) is basically the value for 'pure' black carbon – it's not a realistic RI for atmospheric aerosol (except maybe at the tailpipe of a diesel vehicle). Are the Mie calculations done for both RI's? If so, which RI is depicted in the results in figure 6? Do the error bars reflect the range of the two RIs? Or is some fraction of the particle size distribution assigned to the first RI and the remaining particles assumed to have the second RI? What is the fraction? Is it constant? Is it size dependent?

**We added these sentences in the LOAC description:**

**Mie scattering theory is used for liquid and transparent particles (refractive index = 1.45), and for solid and absorbing particles (refractive index = 2+0.6i), separately. Aerosol extinction values are computed for these two different particle compounds. They represent the range within which the true aerosol extinction of the particle falls in. Finally, the LOAC aerosol extinction at 532 nm is determined by averaging these two extinction values.**

**We also added the following sentence to the Figure 6 caption:**

**Horizontal bars show the standard deviation of the CALIOP data and the LOAC aerosol extinction range of the two refractive indices used in the calculation of the aerosol extinction.**